# Hydrochar Application Improves Growth and Intrinsic Water Use Efficiency of *Populus alba*, Especially during Hot Season

Giovanna Battipaglia [1],*, Francesco Niccoli [1], Jerzy Piotr Kabala [1], Rossana Marzaioli [1], Teresa Di Santo [1], Sandro Strumia [1], Simona Castaldi [1], Milena Petriccione [2], Lucio Zaccariello [1], Daniele Battaglia [1], Maria Laura Mastellone [1], Elio Coppola [1] and Flora Angela Rutigliano [1]

[1]  Department of Environmental, Biological and Pharmaceutical Sciences and Technologies, University of Campania Luigi Vanvitelli, Via Vivaldi 43, 81100 Caserta, Italy

[2]  Council for Agricultural Research and Economics (CREA), Research Centre for Olive, Fruits and Citrus Crops, Via Torrino 3, 81100 Caserta, Italy

*  Correspondence: giovanna.battipaglia@unicampania.it

**Abstract:** Hydrochar, carbon-rich material produced during the thermochemical processing of biomass, is receiving increased attention due to its potential value as soil amendment. It can increase agroforestry systems' productivity through direct and indirect effects on growth and soil quality. Hydrochar may also directly help mitigate climate change by sequestering stable carbon compounds in the soil and perhaps indirectly through increased C uptake by trees. In this research, we aim to evaluate how the application of hydrochar produced by two feedstock types, *Cynara cardunculus* L. (Hc) residuals and sewage sludge (Hs), and in two different doses (3 and 6 $kg\ m^{-2}$) could improve the growth and water use efficiency of *Populus alba* L., a fast-growing tree species largely used in agroforestry as bioenergy crops and in C sequestration. We considered five plants per treatment, and we measured apical growth, secondary growth, leaf area and intrinsic water use efficiency in each plant for the whole growing season from February to October 2022. Our results highlighted that hydrochar applications stimulate the growth and water use efficiency of plants and that the double dose (6 $kg\ m^{-2}$) of both hydrochars, and particularly Hc, had positive effects on plant performance, especially during extremely hot periods. Indeed, the year 2022 was characterized by a heat wave during the summer period, and this condition allowed us to evaluate how plants, growing in soils amended with hydrochar, could perform under climate extremes. Our findings showed that the control plants experienced severe damage in terms of dried stems and dried leaves during summer 2022, while hydrochar applications reduced these effects.

**Keywords:** agroforestry; biowaste; climate change; heat waves; water use efficiency

## 1. Introduction

Global warming negatively impacts plant productivity, development [1] and crop yields [2] in many areas of the globe, inducing important cascade effects on related ecosystems services, including the C sink capacity of terrestrial ecosystems, and posing further challenges to food provisioning in view of the forecasted increase in population [3]. Adaptation strategies and solutions to minimize the negative impacts of climate change and climate extreme on agroforestry systems are a key priority [4–6]. Mitigation strategies include the recycling and valorization of waste material [7]. In the frame of a circular economy policy, with the aim to extend the life cycle of products, in recent years, great attention has been paid to technologies that aim to convert biowaste into high-quality products to be productively used again.

Hydrochar is a carbonaceous material produced from biomass residues using hydrothermal carbonization [8–11]. It is characterized by a low pH, high carbon quantity, high nutrient levels, a good heavy metal absorption capacity and a good carbon availability

for microorganisms in the soil [10]. Recent studies investigated the effects of hydrochar for plant production, including its influence on seed germination, plant morphology, crop productivity or nutrient release from the hydrochar to the soil [12–14]. As a soil amendment, hydrochar enhances the effects of fertilizer on plants' growth and increases the amount of water that can be retained by the soil [15]. However, the actual benefits of these effects on plant performance in response to climate constraints, such as high temperature, have been poorly investigated. In contrast, insights into the possible beneficial effects of the presence of hydrochar in the soil under the interactions of heat waves are essential to limit and/or mitigate the dramatic impact of current climate change on plant health and productivity. In this context, our research aims to verify the effect of different types and doses of hydrochar on the growth and water use efficiency of an ecologically and economically important species such as *Populus alba* L. (white poplar), a fast-growing tree species considered one of the best candidates in the agroforestry for bioenergy production and C sequestration. The growth development of the species was followed during the 2022 growing season, characterized by a summer with an extreme climate period (July) in terms of temperature (www.copernicus.eu accessed on 1 January 2023). The hydrochars produced by sewage sludge (Hs) and by *Cynara cardunculus* L. (Hc) feedstocks were added in two doses (3 and 6 kg m$^{-2}$) to soil where one-year *P. alba* plants were growing. The specific objectives were to (1) assess the impact on growth and intrinsic water use efficiency of the different types and doses of hydrochar in plants growing with it, in comparison to control plants; (2) identify whether the presence of hydrochar could positively or negatively affect plants experiencing high temperature during the growing season.

These objectives will test the hypotheses that: (a) hydrochar effects on plant growth are highly dependent on feedstock properties; (b) increasing application rates will result in a relevant increase in growth and water use efficiency; (c) hydrochar application could mitigate the effect of heat waves and extreme events on plant productivity, increasing plants' water availability. To the best of our knowledge, this is the first study in which the combined effects of hydrochar application with extreme weather events on plant growth and water efficiency are investigated. Moreover, no literature data exist about the actual benefit of hydrochar as amendment soil to mitigate the effects of the studied environmental constraints. Therefore, the expected results will not only provide an innovative contribution to the limited existing body of research focused on the effects of the application rate of hydrochar on plant growth but will also allow us to fill the gaps regarding climate-change-driven impacts on plant fitness.

## 2. Materials and Methods

### 2.1. Experimental Set-Up

The hydrochar used in this experiment was produced according to the procedure described in Zaccariello et al. [16] starting from two feedstocks: *Cynara cardunculus* L. (Hc) and sewage sludge (Hs). Both hydrochars were obtained by operating the pilot-scale hydrothermal carbonization plant at a reaction temperature of 250 °C and a residence time of 6 h.

The C, H and N contents of feedstock and hydrochar were determined using a LECO CHN-S 628 elemental analyzer, the ash content was measured by incineration at 600 °C for 4 h and, in contrast, the oxygen was calculated according to Zaccariello et al. [11]. The hydrochar composition differed on the basis of the feedstock used (Table S1), with C (%) of 30.6 in Hs vs. 61.8 in Hc; H (%) of 5.22 in Hs vs. 6.15 in Hc; N (%) of 0.79 in Hs vs. 1.08 in Hc; ash content (%) of 59.2 in Hs vs. 18.3 in Hc; C/N of 38.7 in Hs vs. 57.3 in Hc. The effects of two hydrochar types (Hc and Hs) applied at two doses (3 and 6 kg m$^{-2}$) were evaluated in a pot experiment and compared to control plants (C, growing without hydrochar). For each treatment, five clones of one-year-old *P. alba* were planted in pots in January 2022 and placed randomly in the CREA nursery center (Council for agricultural research and analysis of the agricultural economy) in Caserta (41.07° N, 14.31° E), Italy. The pots (21 × 16 cm) were filled with 2 kg of soil (depth 0–20 cm) collected in the municipality of Castel Morrone

(Caserta, Italy) and coarsely sieved (<0.4 cm) in order to exclude plant residues and large stones. The texture soil's fine hearth was loamy–sandy (53.6% sand, 32.7% silt and 13.7% clay), moderately acid (pH: 5.86) and nonsaline (electrical conductivity < 2000 $\mu$S cm$^{-1}$), according to Soil Survey Manual (2017), with a water-holding capacity (WHC) of 41.5%, bulk density (BD) of 1.53 g cm$^{-3}$, porosity of 42.3%, cation exchange capacity (CEC) of 14.4 cmol kg$^{-1}$, organic C (C$_{org}$) of 29.4 g kg$^{-1}$ and N of 0.19 g kg$^{-1}$ (Table S2; measurement methods as in [11,17,18]). The plants were monitored for a whole growing season, and the soil was watered by applying 0.5 l per pot daily during the dry season. Temperature conditions were monitored through daily measurements using a Davis Vantage Pro 2 weather station (placed inside the CREA experimental field).

### 2.2. Experimental Measurements

The diameter and height of each clone were measured monthly from February to November 2022. The diameters were measured with an electronic caliper (0.01 mm of precision, RS PRO, Sesto San Giovanni, Milano, Italy), while the heights were measured from the base to the apex of the stem using a measuring tape (1 mm of precision). The surface area of plant leaves was measured each month and for each plant following the direct method described by Chen (1996) [19]. Following a stratified sampling design, the leaves of each plant were classified into three size classes (small, medium, and large) and counted on a monthly basis. At the same time, a sample of five leaves (one per each plant) for each size class was collected and photographed on graph paper in order to measure the leaf area using the ImageJ software and to extend the analyses to the whole leaves' dataset. All the measurements described above were performed by the same operator for the entire duration of the experiment. The growing season was determined by empirical growth and phenological observation [20]. The growth started in February 2022, and it ended in October 2022. Finally, the damage due to heat waves experienced in the summer period was evaluated in terms of the number of dried leaves and stems.

### 2.3. Intrinsic Water Use Efficiency (WUE$_i$)

Each month, the leaves used for leaf area measurement were air dried, pulverized with a pulverizing mill (ZM 1000, Retsch, Haan, Germany) and weighed in a tin capsule for carbon isotopic measurements. The isotopic composition was measured at the IRMS laboratory of the University of Campania "Luigi Vanvitelli" by mass spectrometry with a continuous flow isotope ratio (Delta V Advantage, Thermo Scientific, Waltham, MA, USA). The standard deviation for the repeated analysis of an internal standard was better than 0.1‰. The $\delta^{13}$C series were corrected for the fossil fuel combustion effect [21].

The WUE$_i$ for each treatment was derived from leaves' $\delta^{13}$C values. The WUE$_i$ is defined as the ratio of the rate of carbon assimilation (A, photosynthesis) and stomatal conductance (gs); it can be calculated as:

$$\text{WUE}_i = \text{A/gs} = c_a * [\text{b} - \delta^{13}\text{C}_{atm} - \delta^{13}\text{C}_{leaf})]/[(\text{b} - \text{a}) * 1.6]$$

where $c_a$ is atmospheric CO$_2$-concentration obtained by NOAA (http://www.esrl.noaa.gov/ accessed on 1 January 2023, Mauna Loa station); $\delta^{13}$C$_{atm}$ is the isotopic ratio of atmospheric CO$_2$; $\delta^{13}$C $_{leaf}$ is the isotopic ratio of the leaves; a (4.4‰) is the fractionation due to diffusion; and b (27‰) is the biochemical fractionation [22,23].

### 2.4. Data Analysis

All data were checked for normality before applying inferential statistics. To compare the results of structural analyses of the different groups, a one-way ANOVA was applied considering the five treatments (control; 3 and 6 kg m$^{-2}$ dose of *C. cardunculus* hydrochar, Hc3 and Hc6; 3 and 6 kg m$^{-2}$ doses of sewage sludge hydrochar, Hs3 and Hs6), using Tukey's post hoc test for multiple comparisons between the different treatments. All statistical analyses were performed with R-studio 4.1 software, (Version 1.4.1564, Vienna, Austria) applying a significance threshold of *p* < 0.05.

## 3. Results

### 3.1. Weather Conditions

The experimental area is characterized by a Mediterranean climate with mild, wet winters and warm, hot and dry summers. In the last 5 years, the area has experienced very hot summers, classified as (heat waves www.copernicus.eu accessed on 1 January 2023), and this situation was also foreseen for summer 2022. Indeed, a hot summer occurred, as our working hypothesis expected. During the experimental period (January–October 2022), the meteorological station located at CREA recorded average temperatures of 17.5 °C, with a maximum of 22.9 °C and minimum of 12.4 °C (Figure 1). Although between February and April the temperatures were mild and often below 20 °C, starting from spring, the temperatures gradually increased until they reached significant heat peaks during the summer. July 2022 was found to be the hottest month, classified as a heatwave (www.copernicus.eu, accessed on 1 January 2023). In this period, the maximum temperature was on average 33.7 °C, and it frequently exceeded 30 °C (29 days out of 31). Finally, in the following months, the temperatures decreased but remained above the seasonal average: in autumn, the mean temperature was 21.5 °C, while the maximum was 26.7 °C.

A

| Months | Jan | Feb | Mar | Apr | May | Jun | Jul | Aug | Sep | Oct |
|---|---|---|---|---|---|---|---|---|---|---|
| Average Maximum Temperature (°C) | 13.2 | 15.3 | 15.5 | 19.7 | 26.6 | 32.4 | 33.7 | 32.6 | 26.7 | 25.0 |
| Average Mean Temperature (°C) | 8.7 | 10.1 | 10.3 | 14.4 | 20.6 | 25.8 | 27.6 | 26.2 | 21.6 | 19.2 |
| Average Minimum Temperature (°C) | 4.7 | 5.1 | 5.1 | 9.3 | 15.1 | 19.1 | 21.7 | 20.8 | 17.4 | 14.4 |

B

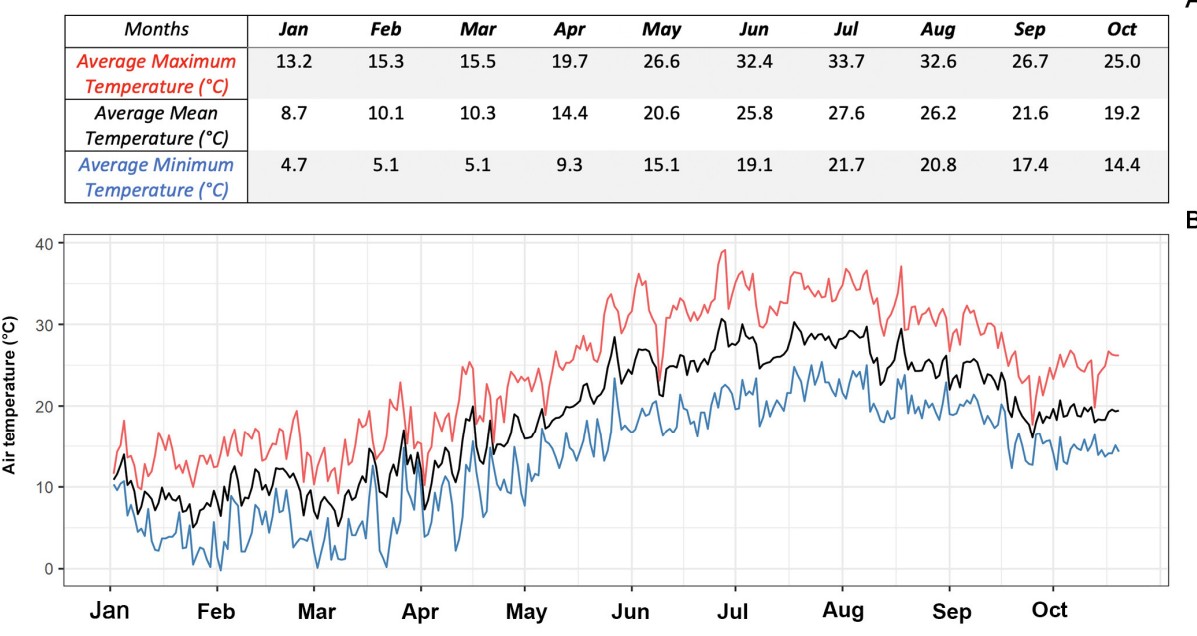

**Figure 1.** (**A**) Table reporting the average minimum, maximum and mean temperature recorded each month. (**B**) Air temperature measured by the weather station installed at CREA during the period of experimental activity. In black: the mean daily values; in red: the maximum; and in blue: the minimum.

### 3.2. Growth Trend

In the first months of the experiment, until May, the white poplar (*P. alba*) plant height did not differ among treatments ($p > 0.05$). Thereafter, a significant difference was only found between Hs6 and Hc6 in comparison to the control (Figure 2A).

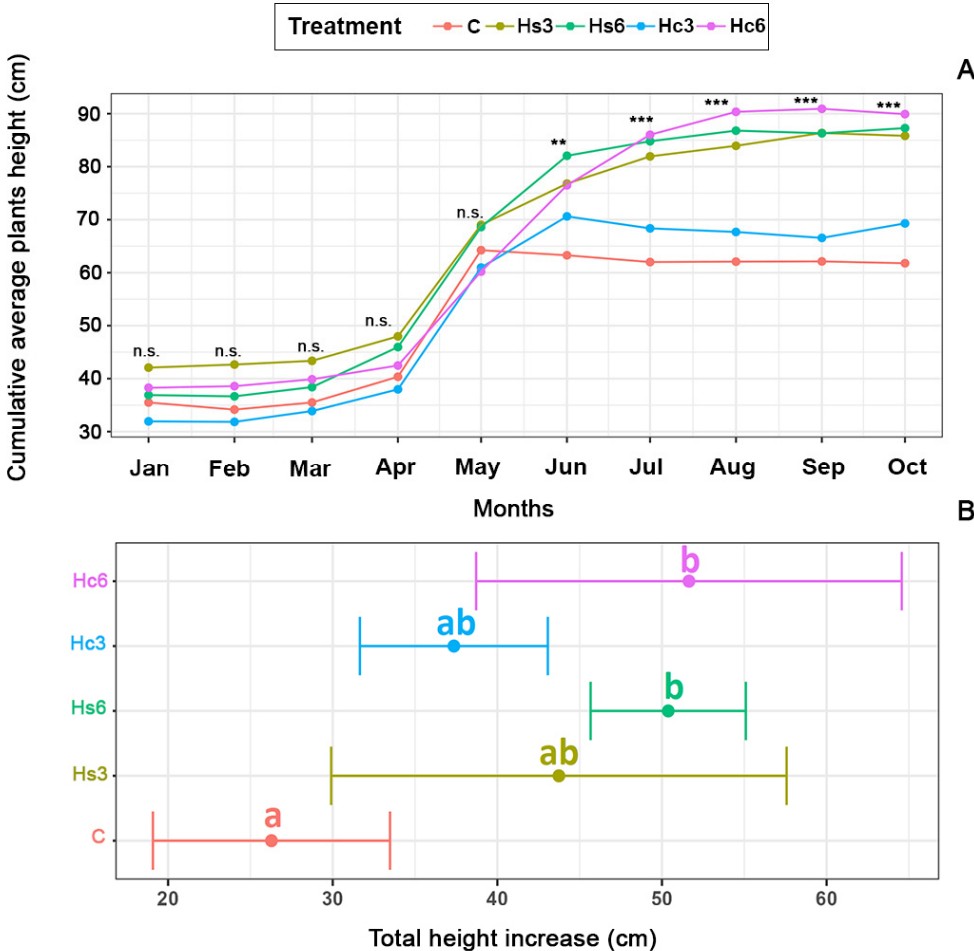

**Figure 2.** (**A**) Cumulative average plant height measured during the experiment with statistical significance (n.s. = not significant, ** $p < 0.01$, *** $p < 0.001$) n = 5. (**B**) Total height increment measured in each treatment during the experiment (bars represent 1 SD). Different letters indicate statistical differences among treatments applying a post hoc Tukey multiple comparison; $p < 0.05$.

Plants treated with the lower dose of hydrochar (Hs3 and Hc3) showed, on average, a higher height compared to control plants, although such a difference was not statistically significant (Figure 2B). A significant ($p < 0.05$) increase in total height was only found when comparing plants growing in Hs6 and Hc6 with control plants (C). While the treated plants grew along the whole season, the control plants had already reached the maximum of their height in the first few months (February–May), stopping their growth in height in the following months.

The same positive effect was found for secondary growth, showing a significant increase in plants growing with Hc6 and Hs6 compared to control plants ($p < 0.05$) starting from July (Figure 3).

During July 2022, extreme temperatures were recorded with maximum temperatures on average of 33.7 °C, and with 29 days out of 31 exceeding 30 °C. The stems and leaves started to dry in the control plants (C) and in the Hc3 treatments (Figure 4).

The percentages of dried stems and leaves in these treatments (80% and 15%, respectively, for the control plants; 83.3% and 32% for the plants that were grown in Hc3) were much higher compared with the other treatments (Figure 4). Plants growing in Hc6 experienced only very limited damage, with less than 1% of dried stems and leaves (Figure 4).

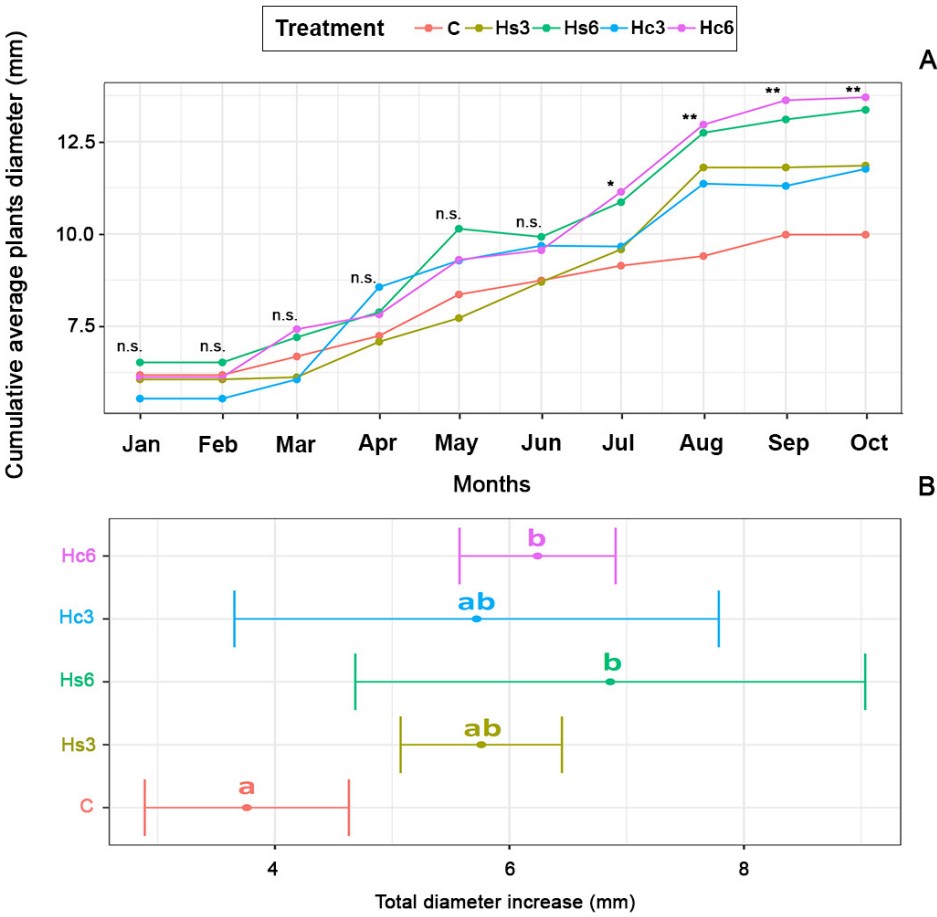

**Figure 3.** (**A**) Cumulative average plant diameter growth measured during the experiment with statistical significance (n.s. = not significant, * *p* < 0.05, ** *p* < 0.01) n = 5. (**B**) Total diameter increment measured in each treatment during the experiment (bars represent 1 SD). Different letters indicate statistical differences among treatments applying a post hoc Tukey multiple comparison; *p* < 0.05.

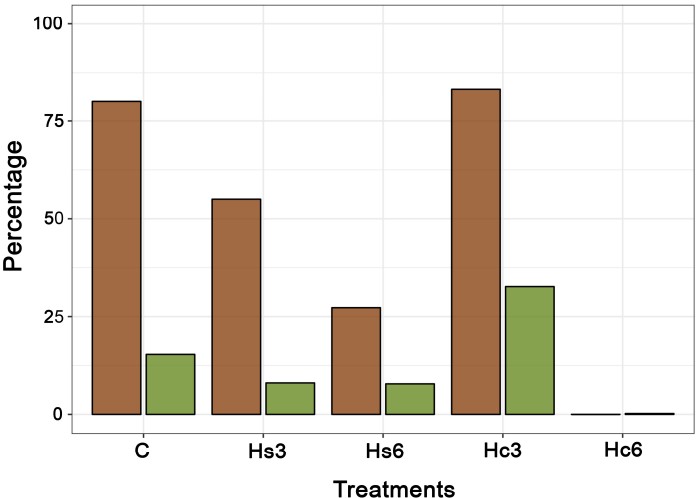

**Figure 4.** Total percentage of dried stems (brown bars) and leaves (green bars) for each treatment, recorded in July 2022.

The effect of high summer temperatures also affected the total leaf area (Figure 5). While in the first months of the experiment, no significant differences were found among

the different treatments, starting from June 2022, the control plants (C) and those growing with 3 kg m$^{-2}$ hydrochar from *C. cardunculus* (Hc3) showed a significant ($p < 0.05$) decrease in total leaf area due to the large number of dried leaves. Therefore, we found a significant difference between treatments (Figure 5).

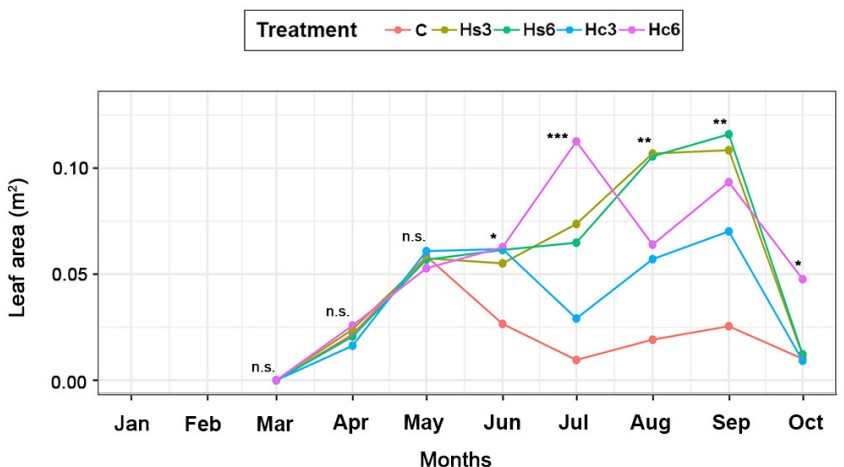

**Figure 5.** Leaf area measurements (average per treatment) during the growing season with statistical significance (n.s. = not significant, * $p < 0.05$, ** $p < 0.01$, *** $p < 0.001$) n = 5.

### 3.3. Intrinsic Water Use Efficiency

No significant differences in values of leaf WUEi were found when comparing the different treatments at the beginning of the growing season, whereas from June onwards, the effect of treatment on $WUE_i$ started to be visible, with the Hc6 treatment reaching the highest intrinsic water use efficiency (Figure 6, $p < 0.05$) and all the hydrochar-treated plants showing higher $WUE_i$ values than in the control, where $WUE_i$ was the lowest throughout the growing season.

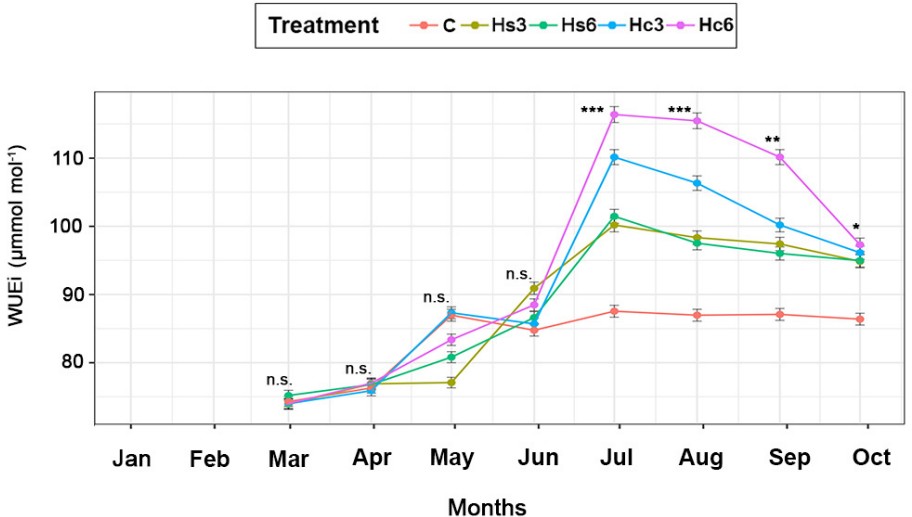

**Figure 6.** Mean $WUE_i$ measured in the leaves of the plants exposed to different treatments in the growing season with statistical significance (n.s. = not significant, * $p < 0.05$, ** $p < 0.01$, *** $p < 0.001$) n = 5, bars indicate 1 standard deviation.

## 4. Discussion

The effect of hydrochar application on plant growth is still under debate, as both positive and negative results have been reported [24–26] depending on the investigated hydrochar feedstocks, which considerably affect the final hydrochar characteristics, and on the

plant species used in the experiments. A previous study, in which hydrochar was applied to several crop species [12], showed that after an initial toxic effect, the nutrients contained in the amendment had positive effects on crops' growth and productivity. Due to their fast growth and global economic significance for industry and energy sectors, hydrochar application on different poplar species was previously investigated, and the positive effect of its application has been reported [27,28]. Baronti et al. [29], using hydrochar produced by maize silage feedstock, reported a consistent increase in the biomass of *P. alba* individuals. Our results on the same species, *P. alba*, are in line with previous findings, providing positive results for hydrochar feedstocks, both plant-based or derived from sewage sludge. Our results showed that none of the hydrochar used induced negative effects on plant growth in terms of height and diameter, suggesting an absence of a phytotoxic effect on the analyzed species [30]. Furthermore, an increase in growth both in terms of height and secondary growth in all the hydrochar applications in comparison to the plants growing without hydrochar (control) throughout the whole experimental period was reported. The types of hydrochar (Hc and Hs) did not significantly affect the height and diameter increase for poplars, even if the feedstocks had different chemical compositions. In particular, Hs had a lower C/N ratio compared to Hc (38.7 vs. 57.3; Table S1). The C/N ratio indicates the amount of carbon relative to the amount of nitrogen present [31]. A lower C/N ratio is often regarded as favorable to plant growth since microorganisms quickly mineralize organic matter and release N available for plant uptake. On the contrary, a higher C/N ratio results in higher microbial immobilization and lower availability for plants [32]. For both feedstocks, the higher dose application presented the best positive effect on plant growth. During the whole growing period, the Hc6 treatment increased the height and the diameter by 96% and 99%, respectively, in comparison to the control plants; in addition, the Hs6 treatment showed a positive effect as it increased the height by 91% and diameter by 80% in comparison to the control plants. The single doses of hydrochar (Hc3 and Hs3) resulted in a positive but not significant increase in terms of height and diameters in comparison to the control plants. Previous studies [30,33] found that the addition of hydrochar at a 5% application rate ($w/w$) was insufficient to improve soil properties essential to plant growth, such as water-holding capacity (WHC) and aggregate stability, and was unable to sustain a long-term fertilization effect. Furthermore, the study of de Jager and Giani [30] demonstrated that the higher application dose resulted in a sustainable increase in the nutrient content ($PO_4$-P and K) and microbial activity, leading to an increase in plant growth. The leaf area data confirmed that double-dose hydrochar applications produced an increase in total leaf area in all the treatments in comparison to the control plants. The positive effect of hydrochar on poplar growth could have also been due to a possible increase in soil pH induced by hydrochar [12,25], notwithstanding the acidic nature of hydrochar, probably determined by microbial activity through proton-consuming reactions [25]. In accordance with these studies, in a parallel experiment carried out on the same soil and using both hydrochar types (Hs and Hc) and doses used in this study (but without plants), a significant ($p < 0.05$) increase in pH, compared to the control, was found 92 days after the hydrochar treatments (Hs3 = 6.61; Hs6 = 6.81; Hc3 = 6.65; Hc6 = 6.69 in treated vs. 5.71 in control), with values significantly higher in the higher dose compared to the lower one. It is well known that *P. alba* grows over a large soil pH range, from acid to strongly alkaline, although optimal growth rates occur at a neutral–alkaline soil pH [34]. Böhlenius et al. [35] demonstrated that the addition of lime in forest acid soil (pH: 5.5–6.5) stimulated the height and diameter in early poplar growth (1–4 year).

Interestingly, the benefit of hydrochar application was very evident during the summer period when a heat wave occurred in the study area and the maximum temperature in July reached 33.7 °C, experiencing temperatures of more than 30 °C for 29 days out of 31. *Populus alba* plants growing with no hydrochar (control) started to show evident signs of desiccation, with 80% having dried stems (see Figure 4). A negative effect was also evident for Hc3, with 83% having dried stems, and Hs3, with 55% having dried stems. The fact that plants growing with Hs6 and Hc6 experienced very limited damage

in terms of dried stems suggests that hydrochar counteracted the negative effects of high temperature on plants and soil. Indeed, high temperature can affect microclimatic variables (which influence seed germination and seedling growth), nutrient availability, soil–water interaction, microbe activity and plant growth [36–38]. An increase in soil temperature could enhance the evaporation rate of soil moisture and decrease the viscosity of soil water. The increased evaporation rates could restrict the movement of water into the soil profile, triggering a scarcity of water to plants [39] and to microorganisms, with consequent reductions in microbial growth [40] and nutrient release in soil through decomposition. In addition, the decrease in water viscosity increases the rate of water absorption in soil and nutrient transport in roots, which hugely affects the photosynthetic activity [41]. It has been demonstrated that soils exposed to high temperatures could be amended with hydrochar to improve soil moisture retention [42]. Hydrochar application could support microbial proliferation and its community structure (by increasing enzymatic activity, biofilm formation and soil aggregate formation) which, together with improved thermal properties of soil, minimize the detrimental impacts of high soil temperatures on soil water retention. Nevertheless, in our study, hydrochar application enhanced intrinsic water use efficiency (WUE$_i$) in plants growing with double-dose hydrochar, and especially in the Hc6 experiment and during the hottest month (July). This result is in line with previous studies reporting that the presence of hydrochar maintained a suitable hydration level for plants during the hottest period, increasing their water use efficiency and nutrient retention capacity [24,29]. This capacity of hydrochar to store water available for plants has been attributed to its highly porous structure, accompanied by a high specific surface area [43], and this has been observed to be different according to the original feedstock carbonization processes [44–46] and application doses [47,48].

It has been demonstrated that the feedstock is essential for the abundance and composition of macropores [49,50].

In plant-derived feedstocks, such as Hc, the macroporosity is generally higher when compared to hydrochar produced from materials meager in lignocelluloses, such as sewage sludge [51]. In addition, porosity positively correlates to O/C ratios with higher O/C (Hc = 0.20 vs. Hs = 0.13) ratios, indicating higher porosity [52].

On the other hand, the differences in soil water content with different amendment rates are due to the significant effect of amendment amounts. This study demonstrated that hydrochar amendment at 3 kg m$^{-2}$ is not enough to significantly impact soil's water-holding capacity, while the double dose results in being efficient, as demonstrated by the increase in WUEi. Moreover, the hydrochar made of *C. cardunculus* feedstock was shown to be significantly more effective than the sewage sludge feedstock during the high-temperature period, probably due to its elemental and structural composition.

## 5. Conclusions

The present study is the first to assess the potential of hydrochar application to mitigate the effect of extremely high temperature on *P. alba* growth. We demonstrated that not only did the hydrochar derived from *C. cardunculus* and sewage sludge not adversely affect the apical and secondary growth of *P. alba* species, but that it also improved plant growth and water use efficiency under extreme weather event. Furthermore, our findings clearly show that feedstock and doses influence plants' responses to hydrochar application, with single-dose application not being sufficient to stimulate plants in terms of growth or intrinsic water use efficiency. The 6 kg m$^{-2}$ dose of Hc and Hs resulted in being a winning application rate, even if these results must also consider plant species as an influential factor, and cognizance must be given to the fact that the same application rate could have very different effects on different plant species.

Considering that heat waves and extreme events have become a worldwide threat to forest plantation sustainability, it is crucial to extend these studies to different species, valuating other environmental constraints and exploiting possible hydrochar application as a measure to increase plant and forest resilience under current and future climate scenarios.

Finally, it is extremely important to promote new multidisciplinary research in which the hydrothermal treatments of feedstocks are coupled with field experiments in order to verify the effectiveness of the produced chars.

**Supplementary Materials:** The following supporting information can be downloaded at: https://www.mdpi.com/article/10.3390/f14040658/s1. Table S1: Mean values (± standard deviations, n=3) of carbon (C), hydrogen (H), nitrogen (N), oxygen (O) and ash contents and C/N ratio of each feedstock (sewage sludge and *Cynara cardunculus* residuals) and hydrochar derived from them (respectively, Hs and Hc). Table S2: Average values (± standard deviation) of water holding capacity (WHC), bulk density (BD), porosity (Po), pH, electrical conductivity (EC), cation exchange capacity (CEC), organic C content ($C_{org}$), total nitrogen (N) of the soil used for the experiments.

**Author Contributions:** G.B. and F.A.R. conceived and designed the experiment; F.N. and J.P.K. performed poplar measurements; L.Z., M.L.M. and D.B. produced and characterized hydrochar; T.D.S., R.M., F.A.R. and E.C., sampled soil and performed soil analyses; S.C., M.P., contributed to the analysis tools; G.B., F.N., J.P.K. and S.S. performed data analyses and data interpretation; G.B. wrote the main part of the manuscript. All authors have read and agreed to the published version of the manuscript.

**Funding:** This research received no external funding.

**Data Availability Statement:** Data Availability Statements are available in the "MDPI Research Data Policies" section at https://www.mdpi.com/ethics, accessed on 24 February 2023.

**Acknowledgments:** The authors wish to thank the VALERE PROJECT of University of Campania "Luigi Vanvitelli" and the MIUR-PRIN 2017 project "The Italian TREETALKER NETWORK: continuous large scale monitoring of tree functional traits and vulnerabilities to climate change".

**Conflicts of Interest:** The authors declare no conflict of interest.

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
