# Peer review of "Hydrochar Application Improves Growth and Intrinsic Water Use Efficiency of Populus alba, Especially during Hot Season"

_forests, doi:10.3390/f14040658_

Round 1
Reviewer 1 Report
This article evaluates how water carbon produced from two feedstock types (Cynara cardunculus L.) residues and sewage sludge improves aspen growth and water use efficiency at two different doses (3 and 6 kg m-2).
Based on measurements of apical growth, secondary growth, leaf area, and intrinsic water use efficiency of five plants per treatment throughout the growing season from February to October 2022, it was found that the application of hydrochar stimulated plant growth and water use efficiency, especially during periods of extreme heat.
Hydrochar is an important functional material that has many applications in agricultural and forestry activities. This article provides a new way to evaluate the application of hydrochar. I suggest that this article could be considered for publication with minor revisions after addressing my following concerns.
1. In contrast to the more substantial data in this paper, there are some errors in the language. I suggest that the authors seek the help of native speakers to further improve the readability of this paper.
2. The authors conducted a large number of effect assessments, such as apical growth, secondary growth, leaf area, and intrinsic water use efficiency. This indicates that hydrochar does work. However, what properties of hydrochar do work needs to be further discussed and illustrated in the context of the relevant literature (e.g., Nat. Commun., 2022, 13, 3616; Green Energy Environ., 2023, DOI: 10.1016/j.gee.2023.01.001; J. Anal. Appl. Pyrolysis, 2022, 166, 105627), which is important to improve the quality of research.
3. The discussion and research on hydrochar are also very important for the mechanism of improvement. The authors may consider adding some related contents to further deepen the study.
Author Response
We thank the review for the positive feedback,
Please find below a detailed point-by-point response to all comments.
- In contrast to the more substantial data in this paper, there are some errors in the language. I suggest that the authors seek the help of native speakers to further improve the readability of this paper.
R1 Done, the manuscript has been revised by an English expert
- The authors conducted a large number of effect assessments, such as apical growth, secondary growth, leaf area, and intrinsic water use efficiency. This indicates that hydrochar does work. However, what properties of hydrochar do work needs to be further discussed and illustrated in the context of the relevant literature (e.g., Nat. Commun., 2022, 13, 3616; Green Energy Environ., 2023, DOI: 10.1016/j.gee.2023.01.001; J. Anal. Appl. Pyrolysis, 2022, 166, 105627), which is important to improve the quality of research.
R2 We thank the reviewer for the comments and for the valuable papers. We added them in our discussion and in references.
3The discussion and research on hydrochar are also very important for the mechanism of improvement. The authors may consider adding some related contents to further deepen the study.
R3 Done, we added in the conclusion the importance of expanding research in the different field from production to utilization of hydrochar.
Reviewer 2 Report
The manuscript is written very well and the discussion was explained in an excellent technical way. However, one crucial thing missing is, it would be more meaningful if the char properties (surface area, pore size, and pore volume) and functional groups present on the char are determined. Therefore, a proper mechanism can be understood for how the plant growth is related to the hydrochar application. Thereby, the authors can confidently draw a conclusion instead of ending the discussion with both positive and negative results, as reported in lines 227 to 229.
Author Response
We thank the review for the positive feedback, and we agree that the hydrochar characterization in terms of porosity and surface area could be an added values. However, this characterization required long and complex methodologies that is not the main aim of this paper, who focus on hydrochar applications on poplar plants. However, we followed referees’ suggestions, and we were able to discuss about the difference in porosity of the two kinds of hydrochar. In particular we argued that It has been demonstrated that the feedstock is essential for the abundance and composition of macropores.
In plant-derived feedstocks, such as Hc, the macroporosity generally is higher when compared to hydrochars produced from materials meager in lignocelluloses such as sewage sludge. Further, porosity positively correlates to O/C ratios with higher O/C (Hc= 0,20 vs Hs= 0,13) ratios indicating higher porosity (Zhu et al., 2015).